# Adaptive Steered Frequency–Wavenumber Analysis for High-Frequency Source Localization in Shallow Water

**DOI:** 10.3390/s25072036

**Published:** 2025-03-25

**Authors:** Y. H. Choi, Gihoon Byun, Donghyeon Kim, J. S. Kim

**Affiliations:** 1Department of Ocean Engineering, Korea Maritime and Ocean University, Busan 49112, Republic of Korea; hwa1470@gmail.com (Y.H.C.);; 2Department of Convergence Study on the Ocean Science and Technology, Korea Maritime and Ocean University, Busan 49112, Republic of Korea; 3Scripps Institution of Oceanography, University of California San Diego, La Jolla, CA 93093, USA; dok054@ucsd.edu

**Keywords:** steered frequency–wavenumber analysis, source localization, sparse vertical line array, adaptive array signal processor

## Abstract

In shallow-water environments, source localization often suffers from reduced performance in conventional array signal processing techniques for frequency bands above 1 kHz due to environmental mismatch. A recently proposed technique, called the steered frequency–wavenumber (SFK) analysis method, overcomes this limitation. By incorporating beam-steering techniques into frequency–wavenumber analysis, this method enables target localization even in sparse conditions where high-frequency signals are received. This study extends the SFK method by applying various adaptive signal processing techniques, with a particular focus on the minimum-variance distortionless response and white noise gain constraint methods. Using snapping shrimp sounds from the SAVEX15 experiment, we analyzed localization performance and compared it with the Bartlett SFK approach. The snapping shrimp signals have frequency components ranging from 5 to 24 kHz and exhibit impulsive characteristics with a duration of 0.2 ms. Signals recorded by a sparse vertical array of 16 sensors, with a 60-m aperture in 100-m shallow water, enabled the localization of a source at a range of 38 m and a depth of 99.8 m.

## 1. Introduction

When conducting marine research, array sensors are deployed to collect and analyze signals, providing valuable information about the ocean. Key information obtained includes target localization, acoustic propagation characteristics of the region, seabed composition, and ocean noise characteristics.

The process of deploying array sensors and acquiring signals in the ocean requires significant effort and time. Due to the complexity of the marine environment, numerous factors must be considered during analysis. Underwater, acoustic waves experience diffraction due to sound speed variations caused by depth changes. Additionally, scattering, reflection, and transmission occur at boundary interfaces, while propagation loss increases with distance. These various factors necessitate extensive analysis to accurately identify targets from received signals [1]. There are many passive acoustic source localization techniques (see [2]). One of the most common techniques, matched field processing (MFP), can be successful at low signal-to-noise ratios (SNRs) when the propagation environment is well known. Additionally, it is a robust source localization technique that can be applied to multipath propagation and data received over long ranges in complex ocean environments.

These MFP techniques continue to be utilized in recent studies. In 2023, Zhu applied the Wasserstein metric to the MFP technique to estimate the location of a source transmitting multiple frequencies. The proposed method reduces ambiguities present in conventional approaches and effectively distinguishes multiple sources [3]. Additionally, in 2023, Jenkins proposed a method for efficiently performing source localization using Bayesian optimization and a Gaussian process (GP) surrogate model. The proposed approach identified the optimal location more quickly and with fewer samples compared to conventional grid search and quasi-random sampling methods [4]. In 2024, Yin introduced an MFP technique that is environmentally and statistically robust based on information geometry. This method mitigates the sensitivity of traditional approaches to environmental uncertainties and enhances performance by incorporating the concepts of Riemannian distance and a modified Jensen–Shannon distance [5].

The MFP method has an upper limit on the source signal frequencies that allow localization when the propagation environment is not perfectly known. In practice, this limit is generally considered to be 1 kHz in shallow-water environments [6]. In this paper, snapping shrimp data near the array in a shallow water environment were utilized to compare various MFP techniques. However, rather than surpassing the general performance of MFP, the comparison was limited to the specific dataset analyzed in this paper. A previous study [7] proposed the steered frequency–wavenumber analysis (SFK) technique, demonstrating that it successfully localized sources using ocean-recorded data with signal frequencies ranging from 5 to 24 kHz. This significantly exceeded the conventional 1 kHz limit.

The SFK method performs beam steering using a weight vector (or replica) before conducting frequency–wavenumber analysis. Frequency–wavenumber analysis allows simultaneous examination of the frequency components of the signal and the direction from which the signal arrives. When visualizing the frequency–wavenumber spectrum of a broadband signal, striations appear. The presence of striations in a frequency band indicates that the signal contains those frequency components, and the slope of the striations represents the direction of arrival of the signal. When beam steering is applied in the same direction as the received signal, the tilted striations in the steered frequency–wavenumber spectrum align vertically at the point where the wavenumber is zero. This feature was utilized to estimate the target’s location. Additionally, to validate the method, comparisons were made with conventional matched field processing (CMFP), adaptive matched field processing (AMFP), and frequency-difference matched field processing (FDMFP) techniques. The results showed that the FDMFP and SFK methods estimated the snapping shrimp’s location at similar positions [7].

Snapping shrimp are small marine creatures, typically measuring 3–5 cm, that inhabit shallow waters less than 60 m deep, where temperatures exceed 11 °C. They form colonies on or near the seabed, often residing in crevices within structures such as rocks, debris, and coral [8,9,10]. A distinguishing characteristic of snapping shrimp is their asymmetrically enlarged claw, which generates intense impulsive sounds, known as snaps. These sounds result from the rapid closure of the claw, creating cavitation bubbles that collapse with a sharp noise [8,11]. When produced collectively by large colonies, these snaps create a continuous crackling sound that dominates the high-frequency (>2 kHz) underwater soundscape in temperate and tropical coastal regions [8,10,12].

The frequency–wavenumber analysis method was first introduced by Capon in 1969 [13]. In 1981, Hinich developed a beamforming technique based on frequency–wavenumber analysis [14], and in 1984, Masry mathematically proved the frequency–wavenumber analysis method for arbitrary array sensors [15]. In 2009, a study was conducted in the field of radio detection and ranging (RADAR) to analyze spatial aliasing effects observed in the frequency–wavenumber spectrum [16]. When spatial aliasing occurs, grating lobes arise during angle estimation, making it challenging to accurately determine the target’s location.

In 2014, Wang conducted a study on localizing multiple sound sources using the frequency–wavenumber analysis method [17]. In 2018, Geoga applied frequency–wavenumber analysis in the field of fluid dynamics to estimate fluid oscillations and transmitted energy [18]. Although extensive research has been conducted on frequency–wavenumber analysis, most studies have focused on signals within frequency bands similar to the design frequency of the array sensors used in experiments [19,20,21,22,23].

However, in this study, the localization performance is insufficient under low-SNR conditions, to the extent that localization fails when the dynamic range is 10 dB (see [7]). The existing SFK method is performed based on the Bartlett processor, so it will be referred to as Bartlett SFK. Therefore, this study aims to improve the performance of the Bartlett SFK method by applying adaptive signal processing techniques, specifically the minimum-variance distortionless response (MVDR) and white noise gain constraint (WNC) methods, and to analyze the impact of white noise gain (WNG) variations on localization performance.

Section 2 covers the theoretical background of the steered frequency–wavenumber analysis method and adaptive signal processing techniques. Section 3 provides details on the SAVEX15 experimental setup and the snapping shrimp signals. Section 4 presents the results from the simulation environment and sea trial data. Finally, Section 5 summarizes the study, discusses the conclusions, and explains the implications of the findings.

## 2. Steered Frequency–Wavenumber Analysis

The frequency–wavenumber analysis technique allows for the simultaneous analysis of both the frequency components of the signal under examination and the direction from which the signal is received. When the frequency–wavenumber spectrum of a broadband signal is visualized, striations appear. The presence of striations in a specific frequency band indicates that the signal contains those frequency components, while the slope of the striations represents the direction from which the signal is received. Within the frequency band of the signal, strong energy can be identified, and the striations are formed along a specific slope, conveying information about the arrival direction of the signal.

However, when a broadband signal with a frequency higher than the design frequency of the array sensors is received, the striations in the frequency–wavenumber spectrum initially form along a specific slope but then transition to the opposite side at the boundary of the wavenumber axis. This phenomenon arises from spatial aliasing, which occurs when array sensors are uniformly spaced based on the design frequency and a high-frequency signal is received. When spatial aliasing occurs, the phase information of the signal within its frequency band is not adequately sampled during array signal processing. As a result, the phase information is misinterpreted, leading to incorrect localization results that make it appear as if the signal originated from a different direction.

To overcome this issue, beam steering is performed before conducting frequency–wavenumber analysis. When the beam is accurately steered toward the signal arrival direction, the striations become vertically aligned at the zero-wavenumber position.

### 2.1. Bartlett SFK

Y.H. Choi et. al. conducted a study on estimating the direction of shrimp sounds from the SAVEX15 experiment by utilizing the fact that the slope of the striations remains constant in the frequency–wavenumber spectrum, even when the striations transition at the wavenumber axis boundary [24]. However, a drawback of this method was that the transition phenomenon prevented the full utilization of the broadband signal’s entire frequency band. To address this issue, beam steering was applied to the received signal as described in Equation (Equation 1) before performing frequency–wavenumber analysis [7].(1)$$\tilde{S}\left(x,z,k,\omega \right)=\int_{-\infty }^{\infty }S\left(r,\omega \right){\bar{w}}_{rep}^{H}\left(x,z,\omega \right)e^{ikr}dr$$ In Equation (Equation 1), $\tilde{S}$ represents the SFK spectrum, *x* represents the range in the searching space, *z* denotes the depth, *k* is the wavenumber, and ω is the angular frequency. *S* is the spectrum in the spatial-frequency domain, r represents the position vector of the linear array, and $w_{rep}$ refers to the beam-steering component, where either the ‘steering vector’ for target bearing estimation or the ‘replica’ containing information about the target location can be used. $\bar{w}$ denotes the normalized vector of w, ${\left(.\right)}^{H}$ represents the conjugate transpose, and boldface denotes a column vector of length *N*. *N* indicates the number of array elements. The replica field $w_{rep}\left(x,z,\omega \right)$ is the simulated field from a point source at the test location that reaches the array. The wavenumber axis was determined using the sensor spacing of the array and was equal to −π/d<k<π/d [14]. Here, *d* is defined as sensor spacing. In this paper, the Nyquist wavenumber (π/d) is approximately 0.84.

### 2.2. MV+SFK Formula

The MVDR processor, also known as the minimum-variance (MV) method, maximum-likelihood method (MLM), or Capon filter [25], is an adaptive technique widely used in MFP. It is particularly effective in suppressing sidelobes [26], leading to a notable improvement in the peak-to-sidelobe ratio compared to Bartlett SFK. However, this advantage comes at the cost of increased sensitivity to environmental mismatches, making it less robust than Bartlett in certain conditions. The MV weight vector can be utilized as the beam-steering vector in the SFK method and can be expressed as follows:(2)$${\tilde{S}}_{MV}\left(x,z,k,\omega \right)=\int_{-\infty }^{\infty }S\left(r,\omega \right){\bar{w}}_{MV}^{H}\left(x,z,\omega \right)e^{ikr}dr$$

Here, $w_{MV}$ is the weight vector obtained through the minimum variance method and is a column vector. To calculate the value of $w_{MV}\left(x,z,\omega \right)$ using the minimum-variance method, the following approach can be adopted:(3)$$w_{MV}\left(x,z,\omega \right)=\frac{K^{-1}\left(\omega \right)w_{rep}\left(x,z,\omega \right)}{w_{rep}^{H}\left(x,z,\omega \right)K^{-1}\left(\omega \right)w_{rep}\left(x,z,\omega \right)}$$

The signal used in this study exhibits impulsive characteristics, and impulse signals typically do not allow for sufficient snapshots. To compute the inverse of the cross-spectral density matrix (CSDM), K must be full-rank, which is typically ensured by increasing the number of snapshots. However, due to the use of impulse signals, the diagonal loading technique was applied. In this paper, diagonal loading was set to 10% of the signal power. Through testing various values, a value was determined that yields localization performance while maintaining similarity to the performance of MVDR.

### 2.3. WNC+SFK Formula

The WNC processor suppresses sidelobes and other sources or interferers while maintaining robustness in common scenarios involving replica data mismatches. The WNC is versatile because of its ability to adjust its behavior (thus resolution) from Bartlett to MV ([27,28]) at the expense of inverting CSDM. To have CSDM invertible, we require *L* > *N* (diagonal loading of CSDM can be used to mitigate this requirement). *L* represents an independent basis of the CSDM. The WNC processor is given by(4)$$w_{WNC}\left(x,z,\omega \right)=\frac{{\left(K+\epsilon I\right)}^{-1}w_{rep}\left(x,z,\omega \right)}{w_{rep}^{H}\left(x,z,\omega \right){\left(K+\epsilon I\right)}^{-1}w_{rep}\left(x,z,\omega \right)}$$ The adaptive weights $w_{WNC}$, which correspond to diagonally loaded MV weights, are derived by solving the following equation:(5)$$\begin{array}{c} \min_{w_{WNC}}w_{WNC}^{H}Kw_{WNC}\;\mathrm{subject}\;\mathrm{to}\\ w_{WNC}^{H}w_{WNC}=1,\\ |{w_{WNC}^{H}w_{WNC}|}^{-1}≧\delta^{2}. \end{array}$$

The WNC imposes an additional quadratic inequality constraint such that(6)$$\delta^{2}≦G_{w}={\left|w_{WNC}^{H}w_{WNC}\right|}^{-1}≦\max\left(G_{w}\right)=1,$$
where $G_{w}$ is WNG whose reciprocal is a measure of sensitivity to tolerance errors [29,30]. The constraint value $\delta^{2}$ must be less than or equal to the maximum possible WNG, $\max(G_{w})=1$, which is attained for the conventional Bartlett weight vector, $w_{rep}$. When $w_{WNC}$ computed in Equation (Equation 4) is applied as the beam-steering vector in the SFK method, it can be expressed as shown in Equation (Equation 7).(7)$${\tilde{S}}_{WNC}\left(x,z,k,\omega \right)=\int_{-\infty }^{\infty }S\left(r,\omega \right){\bar{w}}_{WNC}^{H}\left(x,z,\omega \right)e^{ikr}dr$$

This study analyzes how localization performance varies depending on the WNG. The WNG range is from −6 dB to −2 dB, where performance becomes more similar to Bartlett as it approaches −2 dB and more similar to MV as it approaches −6 dB [31]. The localization performance metric used in this study is the peak-to-sidelobe ratio (PSR), which can be calculated using Equation (Equation 8) [32].(8)$$PSR=\frac{g_{max}-\mu_{sidelobe}}{\sigma_{sidelobe}}$$Here, $g_{max}$ represents the maximum value in the localization result, $\mu_{sidelobe}$ denotes the mean value of the sidelobe region, and $\sigma_{sidelobe}$ is the standard deviation of the sidelobe region. A higher PSR indicates a clearer distinction between the main lobe and sidelobes, signifying superior localization performance.

## 3. Simulation

### 3.1. Geometry

A pre-simulation was run before data analysis to determine applicability, and the environmental conditions were identical to those of SAVEX15. At this stage, a noise-free environment was assumed to assess the applicability of the adaptive signal processing technique, and the tilt of the array was set to 0 degrees. As shown in Figure 1b, a Gauss pulse with a band of 1.5–3.5 kHz was used for the transmission signal, and the sound source is situated at a distance of 130 m and a depth of 100 m. The frequency spectrum of Figure 1b is shown in Figure 1c, and Figure 1d illustrates the received signal in the array.

The signal used for analysis was processed using only the direct wave, as shown in Figure 2a (0.08 s–0.11 s). # indicates the number of the array. The frequency–wavenumber spectrum of the signal received as a direct wave is shown in Figure 2b. Due to the short propagation range, the received signal exhibited the characteristics of a curved wavefront. As a result, in the frequency–wavenumber spectrum, the striations were visualized not as a single slope but as overlapping slopes with similar inclinations.

### 3.2. Bartlett SFK

By applying Equation (Equation 1) using the signal from Figure 2a and the replica computed through the acoustic propagation model, the SFK spectra corresponding to each location are obtained, as shown in Figure 3b–d. At the source location (distance: 130 m; depth: 100 m), the striations become vertically aligned, as illustrated in Figure 3c. Utilizing this characteristic, Figure 3a was visualized by integrating the energy at wavenumber zero across the frequency range. The ambiguity surface in Figure 3a is defined by Equation (Equation 9), with the integration range set to the frequency band of the source signal, from 1.5 kHz to 3.5 kHz.(9)$$P\left(x,z\right)=\int_{\omega_{L}}^{\omega_{H}}\tilde{S}\left(x,z,k{|}_{=0},\omega \right)d\omega$$

In Equation (Equation 9), *P* represents the integral of power in the SFK spectrum ($\tilde{S}$) at a specific location where the values of range (*r*) and water depth (*z*) are known. In Equation (Equation 9), $\tilde{S}$ means using the modeled replica vector, and evaluated at k = 0. In Equation (Equation 9), *r*, *z*, $\omega_{L}$, and $\omega_{H}$ represent the range and depth of the searching space and the wideband signal’s lowest and highest frequencies, respectively. Figure 3b represents the SFK spectrum at a distance of 1 m and a depth of 30 m, while Figure 3d corresponds to a distance of 130 m and a depth of 90 m. The SFK method provides source location information through the ambiguity surface; however, additional localization information can be obtained by observing the vertical alignment of the striations in the SFK spectrum, as shown in Figure 3c.

### 3.3. MV+SFK

Adaptive signal processing techniques develop their formulations based on the beamforming process expressed as $w^{H}d$. However, the method proposed in this study is a beam-steering technique. While beamforming techniques estimate the target location through the coherent summation of array sensors, beam-steering methods only compensate for the transfer function at each array sensor, meaning that summation across the array sensors is not performed. Due to this fundamental difference, it was unclear whether applying the $w_{MV}$ computed in Equation (Equation 3) to the beam-steering process would function correctly. However, by expanding the equation for energy integration at wavenumber zero in the SFK spectrum (Equation (Equation 9)), it was found to be equivalent to $w_{MV}^{H}d$. To verify this, the same conditions as in Figure 3 were used, and the results were visualized using Equation (Equation 2), which applies $w_{MV}$ to the SFK method. The resulting visualization is shown in Figure 4.

Due to the short pulse length of 20 ms in the analyzed signal, it was challenging to satisfy the rank condition of the CSDM. Therefore, the diagonal loading technique was applied, with 10% of the signal power used. Compared to the results in Figure 3a, the results in Figure 4a showed a difference of approximately 25 dB in dynamic range. Since the environment is noise-free, a more detailed analysis will be conducted using additional data. Figure 4b represents the MV+SFK spectrum at a distance of 1 m and a depth of 30 m, Figure 4c corresponds to a distance of 130 m and a depth of 100 m, and Figure 4d represents a distance of 130 m and a depth of 90 m. A key characteristic of the MV+SFK spectrum is that the Bartlett SFK spectrum results are altered at wavenumber zero. This occurs because, at nonzero wavenumbers, the interaction between $w_{MV}$ and the spatial Fourier transform term $e^{ikr}$ prevents $w_{MV}$ from functioning properly. However, the proposed method in this study only utilizes the energy at wavenumber zero, meaning that the improper operation of $w_{MV}$ at nonzero wavenumbers in the MV+SFK spectrum is not an issue. Therefore, to apply an adaptive processor to this method, the adaptive weight vector should be computed and substituted into the beam-steering process.

### 3.4. WNC+SFK

By computing the weight vector of the WNC method in the same manner as the MV+SFK method and applying it to the SFK method using Equation (Equation 7), the resulting visualization is shown in Figure 5. The WNG value was set to −3 dB. Figure 4a represents the ambiguity surface, while Figure 5b–d show the WNC+SFK spectrum at different locations.

Figure 5c shows that, unlike Figure 4c, the striation pattern is more similar to that of the Bartlett SFK spectrum. Additionally, the sidelobes in the ambiguity surface are significantly reduced. To quantify this improvement, performance was evaluated and compared using the PSR metric in Equation (Equation 8). The WNC method enhances processing stability by adjusting the magnitude of the weight vector through appropriate gain control between MV and Bartlett. Generally, the boundary range is known to be between −6 dB and −2 dB [31]. To analyze whether the PSR of WNC is appropriately adjusted between MV and Bartlett, the WNG was varied from −6 dB to −2 dB in 0.5 dB increments.

## 4. SAVEX15 Data Analysis

### 4.1. SAVEX15 Geometry

The SAVEX15 experiment took place in May 2015 in the northeastern East China Sea (ECS) aboard the R/V Onnuri [33]. Its primary objective was to gather acoustic and environmental data to investigate the interplay between oceanographic conditions, underwater acoustics, and communication systems in the region. The experiment site featured a relatively flat sandy seabed at a depth of approximately 100 m. Acoustic transmissions, spanning various frequency bands (0.5–32 kHz), were conducted using both fixed and towed sources, transmitting signals to two moored vertical line arrays (VLAs) over distances ranging from 1 to 10 km. Environmental measurements included the water column sound speed profile, sea surface directional wave conditions, and local wind speed and direction. Throughout the experiment, snapping shrimp sounds were continuously detected. However, for this study, we focus on a specific 0.1-second data segment recorded on Julian Day (JD) 146 (May 26) at 16:51 UTC [33]. A schematic of the experimental setup is provided in Figure 6. The bottom-moored VLA comprised 16 elements, evenly spaced at 3.75 m, covering depths from 25 to 81 m within the approximately 100 m deep water column. The sound speed profile (SSP), shown in Figure 6, was obtained from a CTD (conductivity, temperature, and depth) cast conducted on JD 146. The profile reveals an asymmetrical underwater sound channel with a channel axis at approximately 40 m depth. The sampling frequency for the recordings was set to fs = 100 kHz.

A sample of 100 ms data (47.135 s 47.235 s) was used for the analysis (JD146 16:51). Figure 7a shows the received shrimp sound data at each channel, and if the spectrograms were made using the signal (only direct path, red box in Figure 7a.) received by the receiver of channel 16 close to the sea floor, they are shown in Figure 7b. # indicates the number of the array. The frequency band of the shrimp sound is 5–24 kHz, as can be seen in Figure 7b, and direct paths and surface paths exist in 100 ms data. The design frequency of the array, 200 Hz (sensor spacing was 3.75 m), is sufficiently sparse when compared to the frequency band of shrimp sounds.

### 4.2. Data Analysis

When the direct wave of the received signal is analyzed using the frequency–wavenumber method, numerous striations are formed in the 5–24 kHz range, as shown in Figure 3b. Additionally, Figure 8c reveals that the striations are repeatedly formed with a consistent slope. Figure 8c is a magnified view of the 19.9 kHz to 20.1 kHz section of Figure 8b. In Figure 8a, the direct-path SNR of the snapping shrimp signal is approximately 26.7 dB.

By applying the normalized replica $w_{rep}$, which incorporates the SAVEX15 geometry information into the acoustic propagation model(Bellhop) [34], to Equation (Equation 1) and visualizing the Bartlett SFK results using Equation (Equation 9), the resulting figure is shown in Figure 9.

In Figure 9a, the maximum point of the ambiguity surface is located at a distance of 38.39 m and a depth of 99.8 m. The SFK spectrum at this point is shown in Figure 9d. As observed in Figure 9d, at the estimated source location, the striations become vertically aligned at wavenumber zero. In contrast, Figure 9b,c show that the striations change direction at different slopes. Figure 9b represents the SFK spectrum at a distance of 14 m and a depth of 92 m, while Figure 9c corresponds to a distance of 35 m and a depth of 85 m. These figures illustrate the SFK spectra at locations other than the source location, as the transfer function at these points does not correspond to that of the source. The distance and depth intervals in Figure 9a are set to 0.01 m, considering that snapping shrimp, which inhabit the coastal waters of South Korea, are known to have a body length of approximately 2–4 cm [35]. Applying CMFP, AMFP, and FDMFP techniques to the same dataset revealed that only FDMFP and the SFK method successfully estimated the shrimp’s location [7]. Although the target location was successfully estimated, the sidelobe level difference was approximately 6 dB, with the PSR values of 9.68 dB for the SFK method and 6.65 dB for the FDMFP method. Given that this localization method operates effectively even in sparse environments, an additional adaptive technique was applied to further enhance localization performance. The localization results of FDMFP were compared with those of MV+SFK and WNC+SFK, as shown in Figure 10.

The different frequency components of FDMFP were extracted at intervals of 50 Hz, ranging from 50 Hz to 2 kHz. In the MV method, diagonal loading was applied at 10% of the signal power, and the WNG was set to −3 dB. When estimating the source location using a vertical line array, the results are highly sensitive to the tilt of the array sensors. In this paper, the tilt of the vertical array was set to 0.109 degrees based on previous estimation results using the same dataset [7]. The array tilt was varied from −5 degrees to 5 degrees in 0.1-degree increments to estimate the location of snapping shrimp. The estimated location corresponding to the maximum value was determined by comparing the results of the direct path and the surface path [7]. When estimating the target location using a vertical line array, the tilt of the array is a critical factor, requiring precise calibration. In this paper, the array tilt estimated in a previous study was applied, and an adaptive signal processing technique was additionally incorporated. The localization performance (PSR) for both the simulation results and snapping shrimp data was computed from the ambiguity surfaces of the SFK, FDMFP, and MV+SFK methods. For the WNC+SFK method, the WNG was varied from −6 dB to −2 dB in 0.5 dB increments, and the results are visualized in Figure 11.

Figure 11a presents the results for the simulation, where the PSR values for methods other than WNC+SFK are 2.72 dB (SFK), 1.09 dB (FDMFP), and 7.91 dB (MV+SFK). In Figure 11b, which corresponds to the snapping shrimp data, the PSR values are 9.68 dB (SFK), 6.65 dB (FDMFP), and 10.29 dB (MV+SFK). By comparing the results from Figure 10 and Figure 11, it can be observed that FDMFP, which has a broader focal size, recorded the lowest PSR. When comparing SFK and MV+SFK, the MV+SFK method exhibited a higher PSR due to its relatively lower sidelobe levels. Previous studies have indicated that as WNG approaches −6 dB, the performance of WNC+SFK becomes similar to that of MV. However, in both the simulation environment and the snapping shrimp data, significant differences were observed between MV+SFK and WNC+SFK when WNG was set to −6 dB. This difference arises because the MV method applied diagonal loading at 10% of the signal power. To achieve a higher PSR, the magnitude of the diagonal terms in the CSDM must be reduced. Thus, decreasing the amount of diagonal loading could further improve the PSR of the MV+SFK method. When WNG was set to −3 dB, the WNC+SFK method showed the best PSR performance among the four methods in the simulation. However, in the snapping shrimp data, its performance was similar to that of the FDMFP method. This outcome differs from the results in Figure 10, as certain points in the WNC+SFK method produced high localization values despite an overall lower sidelobe level. The presence of a single high value significantly increases the overall variance, leading to a lower PSR. From Figure 11, it is evident that as WNG approaches −6 dB, the PSR increases. However, in the snapping shrimp data, localization failed when WNG was set to −6 dB and −5.5 dB. Based on the findings of this study, it is concluded that the WNC+SFK method achieves optimal performance when WNG is between −5 dB and −3 dB.

## 5. Conclusions

This study extends the previously developed SFK method by incorporating an adaptive processor. Traditional adaptive processors are designed for beamforming-based methods, whereas the proposed approach applies them to the beam-steering process. If only beam steering had been performed, the adaptive processor would not have functioned properly. However, by integrating the energy at wavenumber zero in the SFK spectrum after beam steering, the issue was resolved, and the method was verified to work correctly. Adaptive processors are highly sensitive to noise and environmental mismatches, making them unsuitable for high-frequency sea trial data. However, since the high-frequency Bartlett SFK method functioned successfully, the adaptive processor was also found to operate correctly. The original SFK method was validated by comparing its performance against CMFP, AMFP, and FDMFP, with only FDMFP and SFK successfully estimating the snapping shrimp’s location. The snapping shrimp data used in this paper have a high SNR of approximately 26.7 dB. Therefore, the location of the snapping shrimp could be estimated using only the conventional SFK method. However, as shown in Figure 10a, the level difference between the main lobe and the sidelobes is within 10 dB, which means that if the signal’s SNR were lower, the localization could fail. To overcome this issue, an adaptive signal processing technique was introduced to reduce the sidelobe levels, and the performance was quantified using PSR values. As a result, the WNC+SFK method improved the PSR by 6.25 dB compared to the Bartlett SFK method when WNG was set to −3 dB. In the snapping shrimp data, however, the PSR decreased at the same WNG setting due to high localization values appearing at certain points. Looking only at PSR values, lower WNG resulted in better performance. However, when comparing ambiguity surfaces as shown in Figure 10, the best results were observed when WNG was between −5 dB and −3 dB. The SFK method has some limitations, as it requires a significant amount of processing time when applied to frequencies higher than the array sensor’s design frequency and is highly sensitive to environmental mismatches caused by high frequencies, requiring precise replica accuracy. However, in environments where these limitations are mitigated, SFK can serve as an effective method for identifying precise targets. It is particularly suitable for cases where the search area is small, and the target size is limited. Additionally, if the signal SNR is poor, an adaptive processor can be used to address this issue.

## Figures and Tables

**Figure 1 sensors-25-02036-f001:**
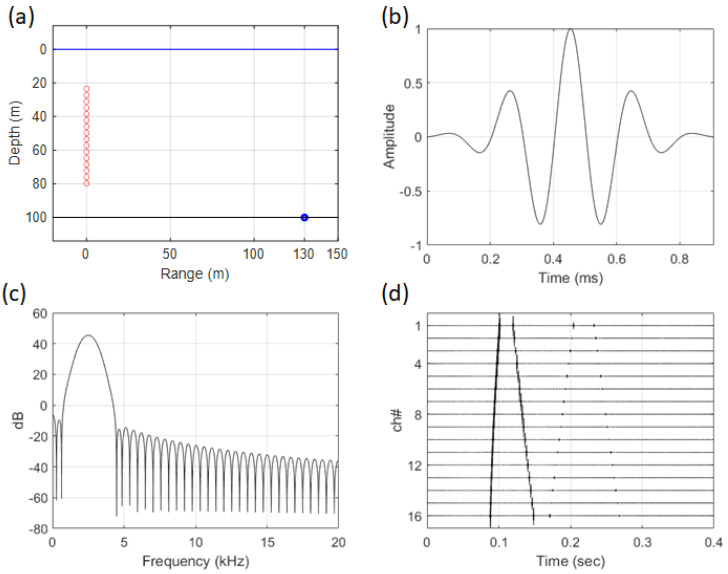
(Color online) (**a**) simulation geometry of array and source, (**b**) Gaussian pulse signal in the time domain, (**c**) spectrum of the source signal, (**d**) received signal in the environment shown in Figure 1a.

**Figure 2 sensors-25-02036-f002:**
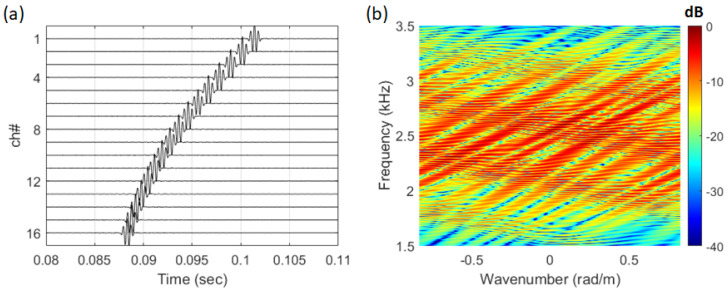
(Color online) (**a**) time series of the direct path, (**b**) frequency–wavenumber spectrum of the direct path signal.

**Figure 3 sensors-25-02036-f003:**
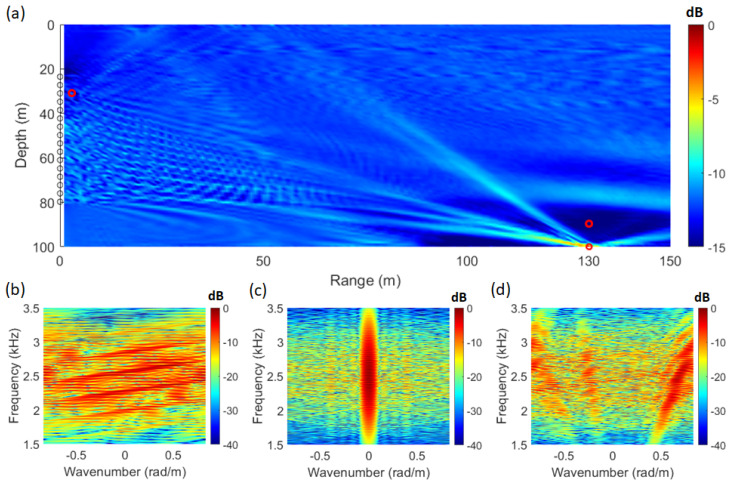
(Color online) (**a**) ambiguity surface from waveguide simulation in the SAVEX15 environment using SFK analysis. (**b**) SFK spectrum at range 1 m; depth 30 m. (**c**) SFK spectrum at range 130 m; depth 100 m. (**d**) SFK spectrum at range 130 m; depth 90 m.

**Figure 4 sensors-25-02036-f004:**
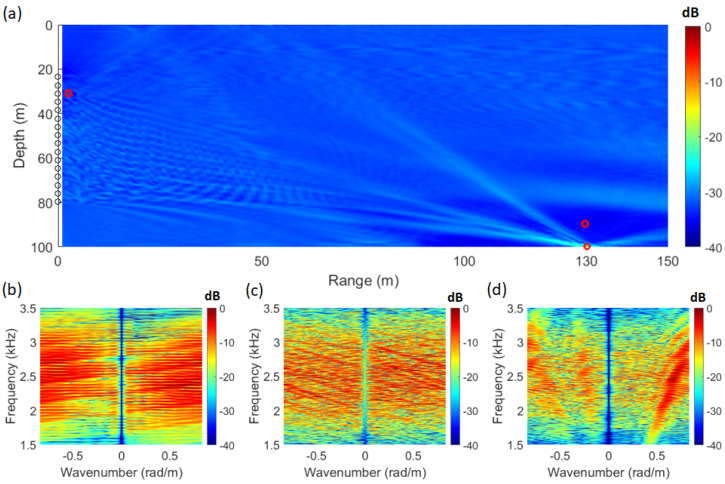
(Color online) (**a**) ambiguity surface from waveguide simulation in the SAVEX15 environment using MV+SFK analysis. (**b**) MV+SFK spectrum at range 1 m, depth 30 m. (**c**) MV+SFK spectrum at range 130 m, depth 100 m. (**d**) MV+SFK spectrum at range 130 m, depth 90 m.

**Figure 5 sensors-25-02036-f005:**
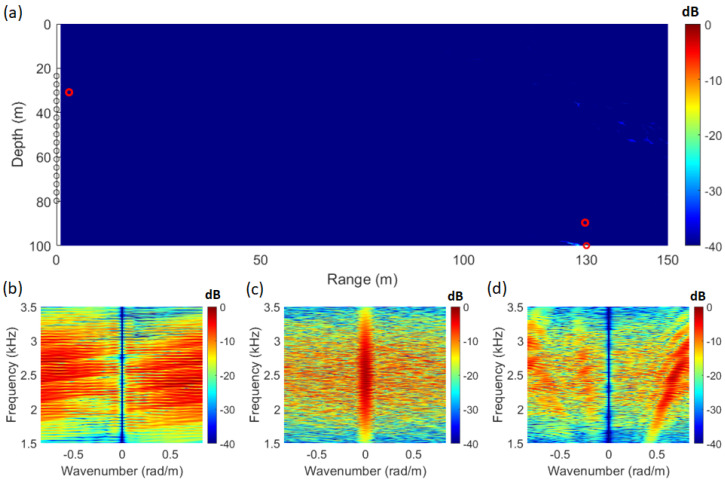
(Color online) (**a**) ambiguity surface from waveguide simulation in the SAVEX15 environment using WNC+SFK analysis. (**b**) WNC+SFK spectrum at range 1 m, depth 30 m, (**c**) WNC+SFK spectrum at range 130 m, depth 100 m, (**d**) WNC+SFK spectrum at range 130 m, depth 90 m.

**Figure 6 sensors-25-02036-f006:**
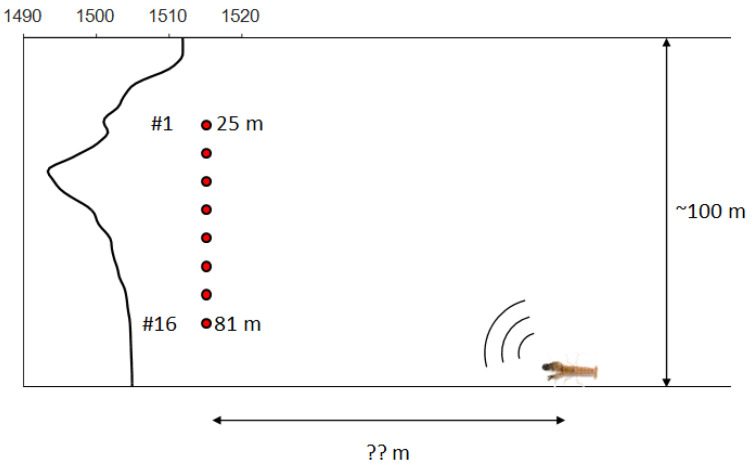
(Color online) the experimental setup of SAVEX15 consists of 16 array sensors deployed at depths ranging from 25 m to 81 m, with a design frequency of 200 Hz. The environment is a shallow-water region with an average depth of approximately 100 m. The sound speed profile corresponds to the average structure of JD146, which exhibits characteristics that form a sound channel.

**Figure 7 sensors-25-02036-f007:**
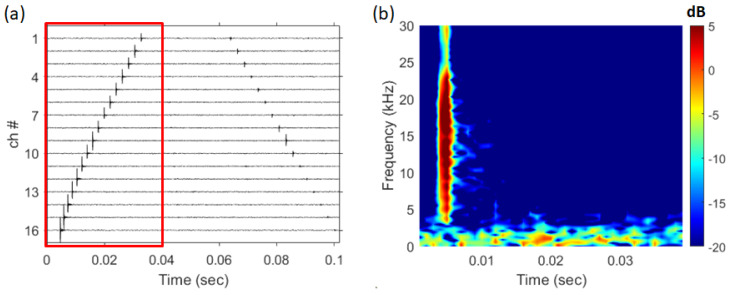
(Color online) (**a**) time series of snapping shrimp signals, (**b**) spectrogram of snapping shrimp signals received on channel 16.

**Figure 8 sensors-25-02036-f008:**
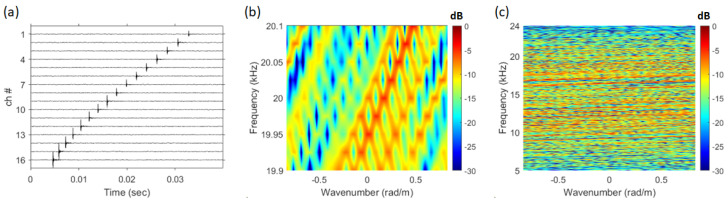
(Color online) (**a**) direct path of the snapping shrimp signal, (**b**) frequency–wavenumber spectrum of Figure 8a (19.9 kHz 20.1 kHz), (**c**) frequency–wavenumber spectrum of Figure 8a (5 kHz 24 kHz).

**Figure 9 sensors-25-02036-f009:**
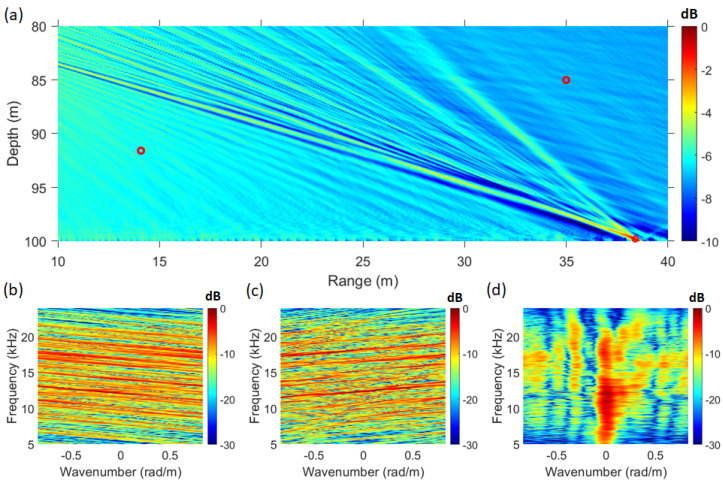
(Color online) (**a**) ambiguity surface from SAVEX15 shrimp data using SFK analysis. (**b**) SFK spectrum at range 14 m, depth 91 m. (**c**) SFK spectrum at (range 38.39 m, depth 99.8 m. (**d**) SFK spectrum at range 35 m, depth 85 m.

**Figure 10 sensors-25-02036-f010:**
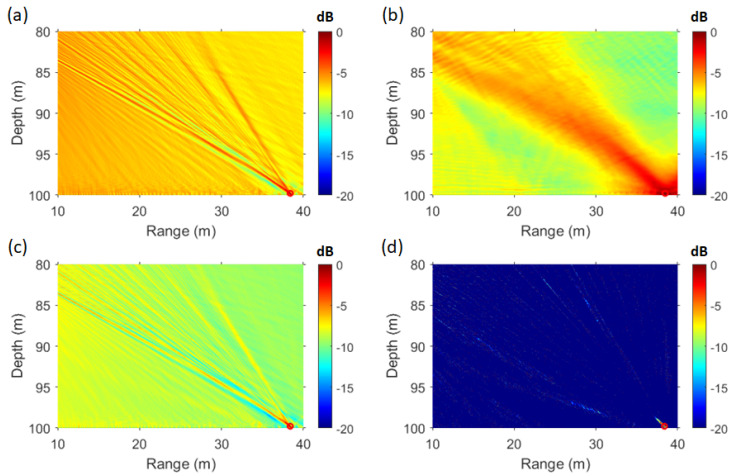
(Color online) source localization results from shrimp data. (**a**) SFK analysis, (**b**) FDMFP, (**c**) MV (10%)+SFK analysis, (**d**) WNC (WNG = −3 dB)+SFK analysis.

**Figure 11 sensors-25-02036-f011:**
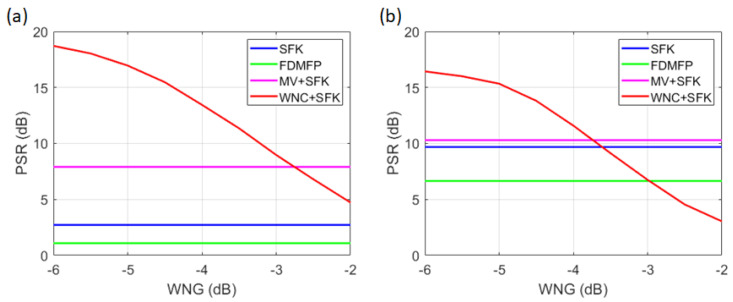
(Color online) PSR results for each localization technique. (**a**) Waveguide simulation, (**b**) SAVEX15 shrimp data.

## Data Availability

Data are contained within the article.

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
