# Peer review of "Adaptive Steered Frequency–Wavenumber Analysis for High-Frequency Source Localization in Shallow Water"

_sensors, 2025, doi:10.3390/s25072036_

Round 1
Reviewer 1 Report
Comments and Suggestions for Authors
This paper proposes an improved adaptive steered frequency-wavenumber (SFK) analysis method aiming at the problem of high-frequency source localization in shallow water environment. It focuses on snapping shrimp noise data recorded in the SAVEX15 experiment. The study tried to improve the performance of the Bartlett SFK method under low SNR conditions. Adaptive signal processing techniques, including minimum variance distortionless response (MVDR) and white noise gain constraint (WNC) methods, are integrated into the SFK framework. The effectiveness of the method is verified by simulation and experimental data and the performance improvement is evaluated by peak-to-sidelobe ratio. Overall, this study introduces innovations in high-frequency source localization, especially in sparse array applications. However, this paper still needs to be improved especially in method description. The comments are listed as follows.
- The paper integrates MVDR and WNC adaptive techniques into the SFK framework, but how many specific improvements does this method have in terms of theoretical or practical advantages compared with the existing high-frequency positioning technology?
- The paper mentions that 10% of the signal power is used for diagonal loading, but the rationale behind this choice and its sensitivity to results are not explained. Similarly, Section 4.2 points out that the array tilt was 0.109 degrees but does not explain how to estimate this value or how its uncertainty affects the positioning performance.
- In Section 2.2, it is mentioned that MVDR is more sensitive to environmental mismatch, but the paper does not deeply discuss how the possible mismatch in SAVEX15 experimental environment, such as the influence of changes in sound velocity distribution or seabed reflection characteristics on the performance of adaptive SFK.
- The introduction and Section 5 claim that the adaptive SFK is designed to improve the positioning performance under low SNR conditions, but this study only tested the noise-free environment in the simulation, and the SAVEX15 data analysis did not explicitly specify the actual SNR value or change.
- Sections 3 and 4 introduce the vertical arrangement of fringes in the frequency wavenumber spectrum as the basis of positioning, but its potential physical mechanism has not been fully explained. Specifically, it is not clear how the arrangement of fringes reflects the propagation path of sound waves or the ability of the array to sample high-frequency signals, especially under sparse conditions.
- There are some inconsistencies in the terminology of the paper, such as the inconsistent use of ' pistol shrimp ' and ' shrimp fishing '. It is recommended to carefully proofread the manuscript to ensure the unity of terminology.
- The abstract should be revised to avoid citing the reference in it.
The paper contains inconsistencies in terminology and the abstract should be revised to avoid citing references. Careful proofreading is recommended to ensure consistency and clarity and the authors should pay attention to grammar and scientific expression.
Author Response
The supplementary file has been attached as additional explanatory material.
Comments 1. The paper integrates MVDR and WNC adaptive techniques into the SFK framework, but how many specific improvements does this method have in terms of theoretical or practical advantages compared with the existing high-frequency positioning technology?
Response 1. Thank you for your valuable comments.
In order to demonstrate theoretical improvements, I introduced the PSR metric. Based on the PSR results, the MV and WNC methods showed performance enhancements. Since MV and WNC optimize the weight vector, it is challenging to mathematically compare them in a complex waveguide environment. However, I introduced the PSR metric to provide a quantitative comparison. Ultimately, in source localization, reducing ambiguity at other locations is desirable. Therefore, PSR was used as a metric to evaluate this aspect.
Comments 2. The paper mentions that 10% of the signal power is used for diagonal loading, but the rationale behind this choice and its sensitivity to results are not explained. Similarly, Section 4.2 points out that the array tilt was 0.109 degrees but does not explain how to estimate this value or how its uncertainty affects the positioning performance.
Response 2. (1) The use of 10% of the signal power for diagonal loading was determined after testing various levels and selecting the value that yielded effective localization performance. I will include this explanation in the main text. (line 145 – 147)
(2) Your observation is absolutely correct. Array tilt is a significant factor in source localization, and it is important to address the associated uncertainties. I will provide a separate file detailing how the array tilt was estimated and will also include a discussion on the impact of uncertainty in the main text. (line 301 – 309)
(PPT Explanation)
(Page 1)
The left figure shows the ambiguity surface obtained by applying the SFK technique to snapping shrimp data. The results were generated by varying the tilt of the vertical line array sensors from -5 degrees to 5 degrees in 0.1-degree increments. In the figure, the red circles indicate the maximum values.
The top-right figure visualizes the depth corresponding to the red circles (maximum values) as a function of the array sensor tilt. The bottom-right figure illustrates the range information as a function of the array sensor tilt.
(Page 2)
Considering the biological characteristics of snapping shrimp, they are assumed to inhabit the seabed at a depth of 100 m. By stacking the seabed slices of the ambiguity surface according to the array sensor tilt, (a) and (b) figures were generated.
Figure (a) represents the stacked results for the direct path, while figure (b) represents the stacked results for the sea surface reflected path. In both (a) and (b), the dotted lines were functionally represented through polynomial fitting. The intersection of these curves was then used to estimate the array sensor tilt.
Comments 3. In Section 2.2, it is mentioned that MVDR is more sensitive to environmental mismatch, but the paper does not deeply discuss how the possible mismatch in SAVEX15 experimental environment, such as the influence of changes in sound velocity distribution or seabed reflection characteristics on the performance of adaptive SFK.
Response 3.
Thank you for your insightful comments. Of course, the adaptive SFK method does not always yield successful results for all datasets, and there were many cases where it failed. As you pointed out, the failures were primarily due to the accuracy of the sound speed profile. Additionally, since this study involves high-frequency signal processing, sensitivity to seabed depth information and roughness also had a significant impact.
Further research with precise experimental data is necessary to analyze these factors in detail. In this paper, the adaptive signal processing technique was applied to the conventional SFK method to demonstrate its potential for extension.
I appreciate your suggestions regarding future research methodologies.
Comments 4. The introduction and Section 5 claim that the adaptive SFK is designed to improve the positioning performance under low SNR conditions, but this study only tested the noise-free environment in the simulation, and the SAVEX15 data analysis did not explicitly specify the actual SNR value or change.
Response 4. (1) In the simulation, a noise-free environment was assumed to assess the feasibility of applying the proposed method. This information will be included in the main text. (lines 175–177)
(2) As you pointed out, the adaptive SFK method was introduced with the intent of improving localization performance in low-SNR environments. However, in reality, the SNR in this study is approximately 26.7 dB, which corresponds to a high-SNR environment. Therefore, even without applying the adaptive signal processing technique, source localization was successfully achieved using the conventional SFK method.
Nevertheless, when visualizing the final results using the conventional SFK method, the level difference between the main lobe and the sidelobes in the ambiguity surface was within 10 dB. To address this issue, the adaptive technique was introduced. To avoid any potential misunderstanding among readers, we will explicitly state the actual SNR values and the objective of the proposed approach in the main text. (lines 276, 348–354)
Comments 5. Sections 3 and 4 introduce the vertical arrangement of fringes in the frequency wavenumber spectrum as the basis of positioning, but its potential physical mechanism has not been fully explained. Specifically, it is not clear how the arrangement of fringes reflects the propagation path of sound waves or the ability of the array to sample high-frequency signals, especially under sparse conditions.
Response 5. (1) The frequency-wavenumber spectrum is obtained through a Fourier transform in the spatial domain, and when examining the Fourier transform equation, it represents the far-field beam output. Therefore, the physical mechanism of the frequency-wavenumber spectrum illustrates the spatial direction and frequency characteristics of the received signal, while the spatial Fourier transform interprets the information related to the signal’s reception location.
(2) The slope of the fringes represents the spatial direction of the signal, and this slope varies depending on the extent of phase compensation. If the phase information at the actual target location is fully compensated, the fringes become vertical, resembling the frequency-wavenumber spectrum of the original signal just before transmission. Additional materials regarding this phenomenon will also be provided.
(PPT Explanation)
(Page 1)
This figure illustrates the geometry for the free-space simulation of the SFK method. The design frequency is 1 kHz, and the number of array elements is 50. The target signals are received from three directions (-60°, 10°, and 70°). The received LFM signals are configured as follows: -60° receives signals in the 5–9 kHz range, 10° in the 1–9 kHz range, and 70° in the 3–6 kHz range.
The top-right figure shows the array configuration, while the bottom-left figure presents the three transmitted signals after applying a Tukey window. The bottom-right figure represents the waterfall time series of the linear array sensors receiving the three signals.
(Page 2)
By performing a frequency-wavenumber analysis on the received signals shown in the bottom-right figure of Page 1, the frequency-wavenumber spectrum is obtained, as displayed in the top-left figure. Since three different frequency bands were received from different directions, three distinct fringe slopes appear. Additionally, because the frequency bands extend beyond the design frequency, aliasing occurs, causing the fringes to wrap around the wavenumber axis.
By applying beam steering to the received signals at -60°, 10°, and 70°, the steered frequency-wavenumber spectrum is obtained. This transformation results in the fringes aligning vertically for their respective arrival directions.
(Page 3)
This page presents additional analysis. By stacking the steered frequency-wavenumber spectra at the wavenumber = 0 for various beam steering angles, a pattern emerges, as shown in the lower figure. This pattern exhibits behavior similar to the grating lobe phenomenon.
(3) It is widely known that the high-frequency signal sampling capability of a sparse array is determined by the spacing between array sensors. In this study, the spacing of the array sensors affects the periodicity of fringe shifts in the frequency-wavenumber spectrum, causing the frequency cycle of repeated axis shifts to shorten.
An animation illustrating this concept will also be provided. The conclusion is that the design frequency of the array corresponds to the shortest cycle at which fringe shifts repeatedly occur, and this phenomenon is most prominent in the endfire direction. As the direction moves toward broadside, the repetition period shortens.
(PPT Explanation)
(Page 1)
A simulation was conducted to explain the principle of fringe formation, and the corresponding geometry is described on this page. The design frequency of the array sensors is 1 kHz, and the signal frequencies were set lower than the design frequency (700 Hz, 500 Hz, and 900 Hz). The arrival directions were set to -60°, 10°, and 70°.
For this simulation, CW signals were used, with 16 array sensor channels and a sampling frequency of 8192 Hz. The bottom-right figure visualizes the transmitted signals.
(Page 2)
The top-left figure presents the time series of signals received from the three directions. The right figure illustrates the received signals for each direction, which were then superimposed to generate the final received signal. The bottom-left figure displays the frequency spectrum of the received signals.
(Page 3)
The received signals were analyzed using a frequency-wavenumber transform, and the resulting frequency-wavenumber spectrum is visualized. The white lines were drawn arbitrarily to illustrate that their slopes indicate the direction of signal arrival.
From this analysis, it can be concluded that the 900 Hz signal arrived from -60°, the 500 Hz signal from 10°, and the 700 Hz signal from 70°.
(Pages 4–6)
To enhance resolution in the frequency-wavenumber spectrum, the number of array sensors was increased from 16 to 50, and the -60° direction was changed to -90°. Additionally, for the signal received at -90°, frequencies were increased from 100 Hz to 4 kHz in 100 Hz increments. The results are visualized in an animation on Page 6.
By observing the animation, it becomes clear how the fringe slopes in the frequency-wavenumber spectrum are formed along the -90° direction.
(Pages 7–8)
If the received signal at -90° is a tonal signal at 2.5 kHz, it becomes ambiguous whether the signal was actually received from -90° or -11.5°. This highlights a key limitation of the SFK method: it requires signals to have a certain bandwidth to be effectively applied.
Comments 6. There are some inconsistencies in the terminology of the paper, such as the inconsistent use of ' pistol shrimp ' and ' shrimp fishing '. It is recommended to carefully proofread the manuscript to ensure the unity of terminology.
Response 6. I will make the necessary revisions to ensure consistency.
Comments 7. The abstract should be revised to avoid citing the reference in it.
Response 7. I will remove the references.
Reviewer 2 Report
Comments and Suggestions for Authors
1. Symbols such as x, r are in confusion. The bold r in Eq.(1) (2) should be explained and the relationship of Symbo r in Eq. (2),x in Eq.(3), range(r) and water depth(z) in Eq.(9) should be explained.
2. The SFK spectrum is also function of test source location, S(x,k,ϖ)?
3. I can’t see anything except threes red circles in figure 4(a). It is better use a reasonable colormap of the ambiguity surface for a detailed analysis.
Author Response
Comments 1. Symbols such as x, r are in confusion. The bold r in Eq.(1) (2) should be explained and the relationship of Symbo r in Eq. (2),x in Eq.(3), range(r) and water depth(z) in Eq.(9) should be explained.
Response 1. Thank you for your valuable comments. I will revise the relevant content accordingly. (Eq. (1, 2, 3, 4, 7, 9), lines 120–122)
Comments 2. The SFK spectrum is also function of test source location, S(x,k,ϖ)?
Response 2. The SFK spectrum used in this study is a function of range ?, depth z, wavenumber ?, and frequency . I have noticed that the current notation may cause confusion for readers. To accurately represent this, it should be written as S(x,z,k,). I will revise the paper accordingly. Thank you. (Eq. (1, 2, 3, 4, 7, 9))
Comments 3. I can’t see anything except threes red circles in figure 4(a). It is better use a reasonable colormap of the ambiguity surface for a detailed analysis.
Response 3. I will redraw the contour by adjusting the dynamic range. Thank you. (Fig. 4(a))
Reviewer 3 Report
Comments and Suggestions for Authors
The paper is devoted to an original technique of underwater sound signal processing, made for localization of sound sources. It allows passive range estimation with vertical line arrays. The present paper is an ongoing study and it expands previous results for sparse arrays. The paper consists of modeling and experimental parts. The paper is well-written however several issues have to be clarified:
- References are not welcome in an abstract of a paper, especially in such form.
- Speculations about MFP in the introduction of the paper are not completely fair. Generally, what is said is true, but usually MFP is applied at larger scales than the examples, presented in the paper. Usually sound source localization with MFP is discussed in context of long ranges with multiple surface and bottom reflections. Even the referred paper [2] deals with 3 km distance while the present paper deals with distances up to 150 m in modeling and around 40 m in the experiment. So the implicit message of the authors that their method overperforms MFP is not yet clear.
- The upper limits of all frequency-wavenumber plots should be discussed. Judging by your array spacing I estimated “Nyquist” k as 0.84. If the most right and most left limits over k are so, then say it to make paper more clear.
Author Response
Comments 1. References are not welcome in an abstract of a paper, especially in such form.
Response 1. I will remove the references from the abstract.
Comments 2. Speculations about MFP in the introduction of the paper are not completely fair. Generally, what is said is true, but usually MFP is applied at larger scales than the examples, presented in the paper. Usually sound source localization with MFP is discussed in context of long ranges with multiple surface and bottom reflections. Even the referred paper [2] deals with 3 km distance while the present paper deals with distances up to 150 m in modeling and around 40 m in the experiment. So the implicit message of the authors that their method overperforms MFP is not yet clear.
Response 2. Thank you for your valuable comments. This paper should focus on performance comparisons based on the data used. To prevent any misunderstanding among readers, I will revise and rewrite the relevant content accordingly. I appreciate your precise feedback. (lines 31–38)
Comments 3. The upper limits of all frequency-wavenumber plots should be discussed. Judging by your array spacing I estimated “Nyquist” k as 0.84. If the most right and most left limits over k are so, then say it to make paper more clear.
Response 3. That is correct. The array spacing used in both the simulation and data is uniformly 200 Hz, resulting in a Nyquist wavenumber k of approximately 0.84. The Nyquist k varies depending on the array spacing, which determines the limit of the wavenumber domain in the frequency-wavenumber spectrum. I will add this explanation to the paper. Thank you. (lines 129–130)